# In vitro and in vivo characterization of SARS-CoV-2 strains resistant to nirmatrelvir

Maki Kiso[1], Yuri Furusawa[1,2], Ryuta Uraki [1,2], Masaki Imai[1,2,3], Seiya Yamayoshi [1,2,3] ✉ & Yoshihiro Kawaoka [1,2,4,5] ✉

Nirmatrelvir, an oral antiviral agent that targets a SARS-CoV-2 main protease (3CLpro), is clinically useful against infection with SARS-CoV-2 including its omicron variants. Since most omicron subvariants have reduced sensitivity to many monoclonal antibody therapies, potential SARS-CoV-2 resistance to nirmatrelvir is a major public health concern. Several amino acid substitutions have been identified as being responsible for reduced susceptibility to nirmatrelvir. Among them, we selected L50F/E166V and L50F/E166A/L167F in the 3CLpro because these combinations of substitutions are unlikely to affect virus fitness. We prepared and characterized delta variants possessing Nsp5-L50F/E166V and Nsp5-L50F/E166A/L167F. Both mutant viruses showed decreased susceptibility to nirmatrelvir and their growth in VeroE6/TMPRSS2 cells was delayed. Both mutant viruses showed attenuated phenotypes in a male hamster infection model, maintained airborne transmissibility, and were outcompeted by wild-type virus in co-infection experiments in the absence of nirmatrelvir, but less so in the presence of the drug. These results suggest that viruses possessing Nsp5-L50F/E166V and Nsp5-L50F/E166A/L167F do not become dominant in nature. However, it is important to closely monitor the emergence of nirmatrelvir-resistant SARS-CoV-2 variants because resistant viruses with additional compensatory mutations could emerge, outcompete the wild-type virus, and become dominant.

Severe acute respiratory syndrome coronavirus 2 (SARS-CoV-2), the cause of COVID-19, is still prevalent three years since its emergence and continues to spread around the world. It accumulates amino acid substitutions frequently and many variants of concern (VOCs) have appeared and caused several waves of infection. Among these variants, omicron (lineage B.1.1.529) was identified at the end of 2021 and became globally dominant[1]. Omicron variants possess more than 30 amino acid substitutions in the spike protein, and its subvariants BA.4/5, BA.4.6, and BA.2.75 are less susceptible to therapeutic monoclonal antibodies including REGEN-COV [casirivimab (REGN10933) plus

imdevimab (REGN10987)], bamlanivimab (LY-CoV555) plus etesevimab (LY-CoV016), evusheld [tixagevimab (COV2-2196) plus cilgavimab (COV2-2130)], and xevudy [sotrovimab (S309)][2–6]. BA.4/5, BA.4.6, and BA.2.75 are susceptible to bebtelovimab (LY-CoV1404) but the latest subvariants BQ.1.1 and XBB escape from it[7]. The antiviral compounds nirmatrelvir, remdesivir, molnupiravir, and ensitrelvir, which target virus proteins other than the spike protein, are effective against such variants and subvariants.

Several antivirals have been widely used for to treat COVID-19 including GS-441524 (remdesivir), EIDD-1931 (molnupiravir),

[1]Division of Virology, Institute of Medical Science, University of Tokyo, Tokyo, Japan. [2]The Research Center for Global Viral Diseases, National Center for Global Health and Medicine Research Institute, Tokyo, Japan. [3]International Research Center for Infectious Diseases, Institute of Medical Science, University of Tokyo, Tokyo, Japan. [4]The University of Tokyo Pandemic Preparedness, Infection and Advanced Research Center, Tokyo, Japan. [5]Department of Pathobiological Sciences, School of Veterinary Medicine, University of Wisconsin–Madison, Madison, USA. ✉e-mail: yamayo@ims.u-tokyo.ac.jp; yoshihiro.kawaoka@wisc.edu

PF-07321332 (nirmatrelvir), and S-217622 (ensitrelvir). Remdesivir and molnupiravir inhibit virus RNA-dependent RNA polymerase, whereas nirmatrelvir and ensitrelvir interfere with 3CL protease [3CLpro; also known as main protease (Mpro) and nonstructural protein 5 (nsp5)]. As these drugs are increasingly used in clinical settings, there is concern about the emergence of viruses with reduced sensitivity or resistance. Amino acid substitutions associated with resistance were reported for remdesivir[8,9] and nirmatrelvir[10,11].

Nirmatrelvir (PF-07321332) is co-formulated with the pharmacokinetic enhancer ritonavir (the co-formulated product is called paxlovid)[12]. When treatment is initiated within three days of onset, approximately 90% protection against severe COVID-19 and hospitalization has been reported[13]. Amino acid substitutions that are responsible for resistance to nirmatrelvir have been reported in 3CLpro (nsp5). In silico analysis showed that the N142L, E166M, Q189E, Q189I, and Q192T substitutions reduced the potency of nirmatrelvir in vitro[14]. A screen using a VSV-based system revealed that the Y54C, G138S, L167F, and Q192R substitutions confer resistance to nirmatrelvir[15]. Naturally occurring amino acid substitutions such as S144A, A173V, and E166D/G also contribute to resistance to nirmatrelvir[10,11,16]. Deep mutational analysis has shown that the E166V, P252L, and T304I substitutions reduce the $IC_{50}$ value to nirmatrelvir[17,18]. In another study, resistant viruses possessing the E166V and T304I substitutions in combination with other amino acid substitutions in nsp5 were selected through virus passaging in the presence of nirmatrelvir in cell culture[19]. The E166V and T304I substitutions conferred strong resistance but resulted in a loss of viral fitness; viral fitness was restored by compensatory substitutions such as L50F and T21I[18]. The L50F and T21I substitutions had little effect on the sensitivity to nirmatrelvir[18]. Another resistant virus selection experiment in vitro identified the combination of the L50F, E166A, and L167F substitutions as being associated with resistance to nirmatrelvir[20]. Each E166A and L167F substitution conferred reduced sensitivity to nirmatrelvir[20]. Recent studies have reported that the virus fitness of viruses possessing the L50F and E166V substitutions and the L50F, E166A, and L167F substitutions is comparable to that of wild-type virus in cell culture[19,20]. Another preprint reported that a recombinant virus possessing the L50F, E166A, and L167F substitutions showed similar pathogenicity in hamsters and transmitted to co-housed hamsters[21]. Therefore, among the many resistant mutations reported, we chose to study Nsp5-L50F/E166V[19] and Nsp5-L50F/E166A/L167F[20] because they appeared unlikely to affect virus fitness in vitro. By using reverse-genetics based on the delta variant, we prepared mutant viruses possessing these substitution combinations and characterized their replication in vitro, their pathogenicity in hamsters, and their airborne transmissibility in hamsters. We also assessed their replication in co-infection experiments with wild-type virus in the presence or absence of nirmatrelvir in the hamster model.

## Results

### Sensitivity of mutant SARS-CoV-2 viruses to antiviral compounds

To characterize mutant viruses possessing amino acids mutations that are responsible for reduced sensitivity to nirmatrelvir, we prepared a wild-type delta (B.1.617.2) variant and its mutant viruses possessing the L50F and E166V substitutions in Nsp5 (Nsp5-L50F/E166V) or the L50F, E166A, and L167F substitutions (Nsp5-L50F/E166A/L167F) by using a BAC-based reverse genetics system. No nucleotide mixtures of more than 10% were identified in the three stock viruses. These three viruses were tested for sensitivity to the antivirals PF-07321332 (nirmatrelvir), EIDD-1931 (molnupiravir), GS-441524 (remdesivir), and S-217622 (ensitrelvir) in a focus reduction assay (Fig. 1a and 1b). The $IC_{50}$ value of PF-07321332 (nirmatrelvir) against wild-type virus was 8.08 μM (95% CI, 5.81–11.2 μM), whereas those against Nsp5-L50F/E166V and Nsp5-L50F/E166A/L167F were greater than 100 μM. Although EIDD-1931 (molnupiravir) and GS-441524 (remdesivir) showed similar $IC_{50}$ values

against all three viruses tested, the sensitivity of Nsp5-L50F/E166V to S-217622 (ensitrelvir) was moderately reduced and that of Nsp5-L50F/E166A/L167F was appreciably reduced; the $IC_{50}$ values were 1.41 μM (95% CI, 0.90–2.20 μM) and 19.34 μM (95% CI, 14.00–26.68 μM). These results indicate that both mutant viruses have reduced sensitivity to PF-07321332 (nirmatrelvir) and show reduced sensitivity to S-217622 (ensitrelvir) as well.

### Propagation of mutant viruses in the presence or absence of nirmatrelvir in vitro

We evaluated the replicative ability of Nsp5-L50F/E166V and Nsp5-L50F/E166A/L167F in vitro. VeroE6/TMPRSS2 cells were infected with each virus at an MOI of 0.001 and virus titers were determined at the indicated timepoints (Fig. 2). In the absence of nirmatrelvir, the growth of Nsp5-L50F/E166V and Nsp5-L50F/E166A/L167F was significantly delayed compared with that of wild-type virus, but the maximum virus titers were similar (Fig. 2a). In the presence of nirmatrelvir, wild-type virus grew poorly, reaching only $3.8 \times 10^4$ PFU at 72 h post-infection, whereas Nsp5-L50F/E166V and Nsp5-L50F/E166A/L167F replicated similarly to in the absence of nirmatrelvir (Fig. 2b). These results indicate that these amino acid substitutions that are responsible for resistance to nirmatrelvir decrease the efficiency of virus replication.

### Pathogenicity of mutant viruses in hamsters

Next, we compared the pathogenicity of Nsp5-L50F/E166V and Nsp5-L50F/E166A/L167F with that of wild-type virus. Hamsters were infected with these viruses and body weight, respiratory functions, and virus titers in the nasal turbinate and lungs were measured at each timepoint. Mock-infected hamsters gradually gained body weight, whereas hamsters infected with wild-type virus lost body weight until 7 days post-infection (Fig. 3a). The body weight of the hamsters infected with Nsp5-L50F/E166V did not change until 6 days post-infection and then gradually increased. Infection with Nsp5-L50F/E166A/L167F did not affect body weight, resulting in a similar body weight change as that seen with mock-infected hamsters.

The respiratory function of the hamsters was assessed by measuring Penh and Rpef, which are surrogate markers for bronchoconstriction and airway obstruction, respectively, by using a whole-body plethysmography system. Wild-type virus significantly impaired Penh and Rpef at 5 dpi, whereas both mutant viruses moderately impaired in Penh and Rpef (Fig. 3b).

We next compared virus titers in the nasal turbinates and lungs of infected hamsters at 3 and 6 days post-infection. At 3 days post-infection, wild-type virus and both mutant viruses replicated in the nasal turbinates to a similar level, whereas the virus titers in the lungs of hamsters infected with Nsp5-L50F/E166V or Nsp5-L50F/E166A/L167F were significantly lower than those of wild-type virus (Fig. 3c). At 6 days post-infection, virus titers in the nasal turbinates of the mutant virus-infected groups were lower than those in the wild-type virus-infected group, although no significant difference in lung virus titer was found among the three groups.

Taken together, these data suggest that Nsp5-L50F/E166V and Nsp5-L50F/E166A/L167F are slightly attenuated compared with wild-type virus.

### Transmission of Nsp5-L50F/E166V and Nsp5-L50F/E166A/L167F between hamsters

We examined whether Nsp5-L50F/E166V and Nsp5-L50F/E166A/L167F maintain airborne transmissibility in hamsters. Naïve hamsters were exposed to infected hamsters at one day after infection and virus titers were measured in the nasal turbinates and lungs of the infected and exposed hamsters at 4 days after infection or 3 days after exposure. The infected and exposed hamsters were separated by a double-layer partition to avoid direct contact. Among the infected hamsters, the virus titers for the wild-type, Nsp5-L50F/E166V, and Nsp5-L50F/E166A/

 

**a**

| Virus | IC$_{50}$ values (µM) (95% confidence interval) | | | |
|---|---|---|---|---|
| | PF-07321332 (Nirmatrelvir) | EIDD-1931 (Molnupiravir) | GS-441524 (Remdesivir) | S-217622 (Ensitrelvir) |
| **Wild-type** | 8.08 (5.81–11.2) | 3.34 (2.38–4.72) | 0.81 (0.66–0.98) | 0.18 (0.14–0.24) |
| **Nsp5-L50F/E166V** | >100 (n.a.) | 5.05 (4.15–6.13) | 1.05 (0.81–1.36) | 1.41 (0.90–2.20) |
| **Nsp5-L50F/E166A/L167F** | >100 (n.a.) | 2.16 (1.50–3.11) | 1.19 (0.95–1.48) | 19.34 (14.00–26.68) |

**b**

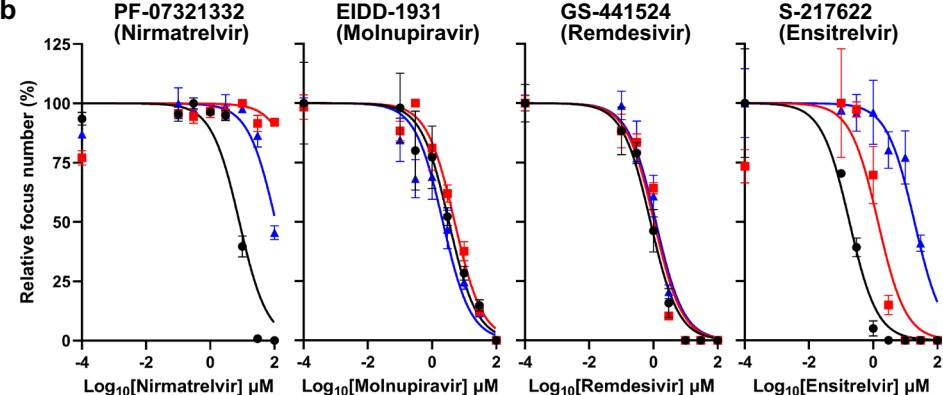

**Fig. 1 | Sensitivity of Nsp5-L50F/E166V and Nsp5-L50F/E166A/L167F to antivirals.** A focus reduction assay was performed to measure the IC$_{50}$ values of PF-07321332 (nirmatrelvir), EIDD-1931 (molnupiravir), GS-441524 (remdesivir), and S-217622 (ensitrelvir) against wild-type and mutant viruses. **a** The IC$_{50}$ values and 95% confidence intervals were calculated based on the inhibitory dose-response curve shown in (**b**). **b** Nonlinear fitting curves of inhibitory dose-response. The data shown are mean virus titers ± standard deviation ($n = 3$ independent experiments).

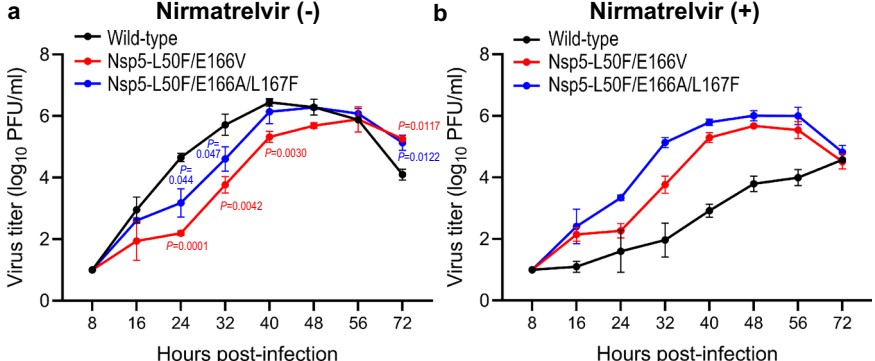

**Fig. 2 | Growth kinetics of Nsp5-L50F/E166V and Nsp5-L50F/E166A/L167F in vitro.** The indicated viruses were inoculated into VeroE6/TMPRSS2 cells at an MOI of 0.001. After 8, 16, 24, 32, 40, 48, 56, and 72 h incubation in the absence (**a**) or presence (**b**) of 20 µM nirmatrelvir, virus titers were determined by use of plaque assays on VeroE6/TMPRSS2 cells. The data shown are mean virus titers ± standard deviation ($n = 3$ independent experiments). Data were analyzed by using a two-way ANOVA followed by Tukey's multiple comparisons.

L167F were higher in the lungs than in the nasal turbinates (Fig. 4, infected). For the exposed hamsters, wild-type virus was detected in 100% (5/5) and 80% (4/5) of the nasal turbinates and lungs, respectively, indicating that wild-type virus was transmitted to all pairs tested (Fig. 4, exposed). Nsp5-L50F/E166V was detected in 60% (3/5) and 20% (1/5) of the nasal turbinates and lungs, respectively, whereas Nsp5-L50F/E166A/L167F was detected in 80% (4/5) of both the nasal turbinates and lungs. The transmission efficiency was not statistically different among the wild-type (100%), Nsp5-L50F/E166V (60%), and Nsp5-L50F/E166A/L167F (80%). These results suggest that Nsp5-L50F/E166V and Nsp5-L50F/E166A/L167F maintain airborne transmissibility in the hamster model, although their efficiency may be slightly reduced.

**Competitive growth of mutant and wild-type viruses in hamsters**
We next examined the competitive growth capability of each virus by co-infecting the wild-type virus with a mutant virus at a ratio of 3:7 based on virus infectious titers. The ratios of each mutant virus to wild-type virus in the inoculum based on deep sequence analysis were 48:52 (wild-type: Nsp5-L50F/E166V) and 42:58 (wild-type: Nsp5-L50F/E166A/L167F) (top panels of Fig. 5). At 4 days post-infection, the frequency of each virus in the nasal turbinates and lungs of the co-infected hamsters was determined. Wild-type virus clearly dominated in the lungs and nasal turbinates of all five hamsters co-infected with wild-type virus and Nsp5-L50F/E166V. Similarly, the proportion of Nsp5-L50F/E166A/L167F relative to the wild-type virus was markedly decreased in the

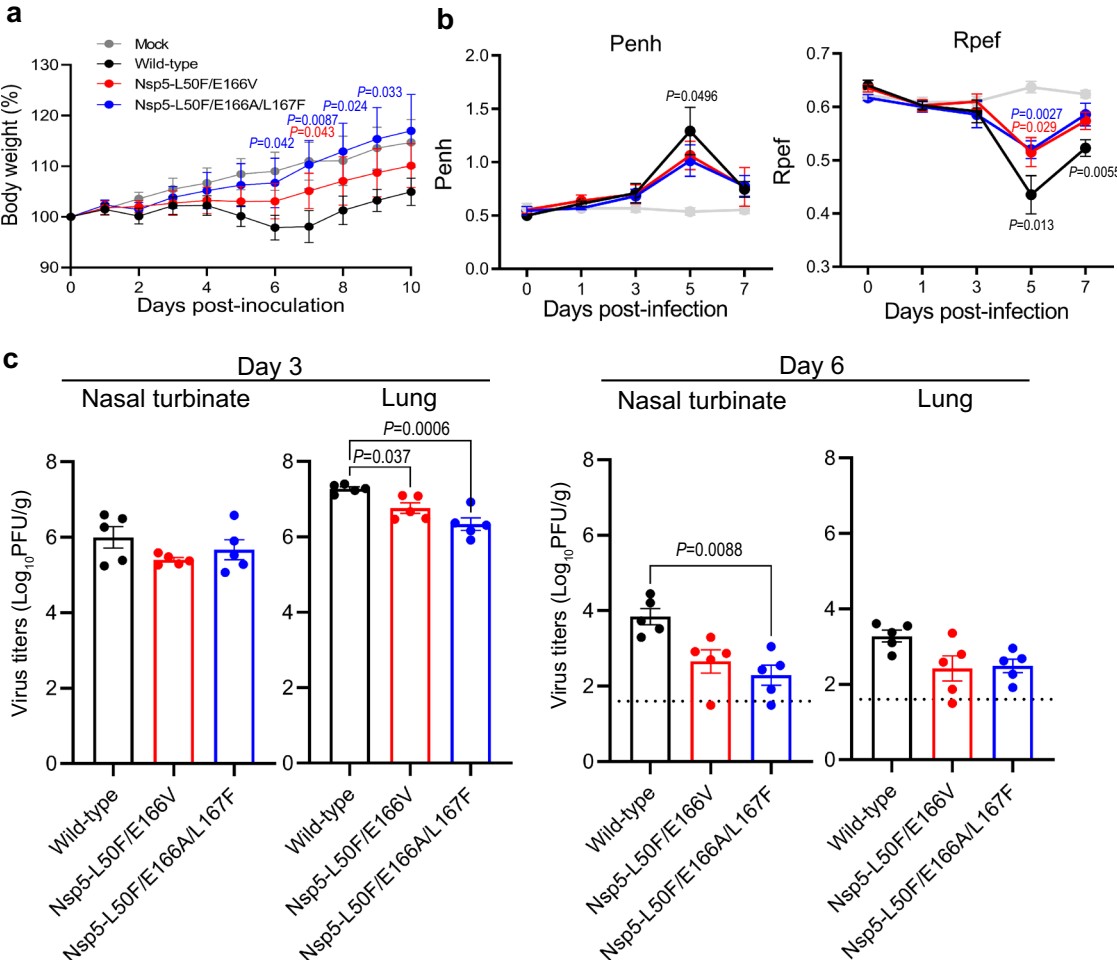

**Fig. 3 | Pathogenicity of Nsp5-L50F/E166V and Nsp5-L50F/E166A/L167F in vivo.**
**a**, **b** Hamsters were intranasally inoculated with $10^5$ PFU of the indicated viruses or with phosphate-buffered saline (mock). Body weight (**a**) and respiratory functions (Penh and Rpef) (**b**) of virus-infected ($n = 5$) and mock-infected hamsters ($n = 5$) were measured daily for 10 days and by using whole-body plethysmography. Data are presented as the mean ± standard deviation and were analyzed by using a two-way ANOVA followed by Tukey's multiple comparisons. **c** Virus propagation in the

nasal turbinate and lungs of hamsters. Hamsters ($n = 10$) were intranasally inoculated with $10^5$ PFU of the indicated viruses and the nasal turbinates and lungs were collected at 3 and 6 dpi for virus titration ($n = 5$ per day). Virus titers were determined by use of plaque assays with VeroE6/TMPRSS2 cells. Points indicate data from individual hamsters and bars show the mean ± standard deviation. The lower limit of detection is indicated by the horizontal dashed line. Data were analyzed by using a one-way ANOVA followed by Tukey's multiple comparisons.

nasal turbinates and lungs of all five animals, although Nsp5-L50F/E166A/L167F was present at 10 and 18% in the nasal turbinates of two hamsters. Next, we performed a coinfection experiment in hamsters treated with nirmatrelvir. The ratios of each mutant virus to wild-type virus in the inoculum were 45:55 (wild-type: Nsp5-L50F/E166V) and 43:57 (wild-type: Nsp5-L50F/E166A/L167F) (bottom panels of Fig. 5). Although the proportion of Nsp5-L50F/E166V or Nsp5-L50F/E166A/L167F was significantly increased in the nasal turbinates ($p = 0.00794$ or $p = 0.00794$) and lungs (($p = 0.00794$ or $p = 0.0476$) of the treated hamsters compared to the untreated animals, the wild-type virus was predominant in most of the hamster nasal turbinates. However, in the lungs of three hamsters, Nsp5-L50F/E166V became dominant, and in the lungs of one hamster, Nsp5-L50F/E166A/L167F became dominant. These results demonstrate that although Nsp5-L50F/E166V and Nsp5-L50F/E166A/L167F are less fit than the wild-type virus, Nsp5-L50F/E166V could be problematic by dominating the wild-type virus in lungs in the presence of nirmatrelvir.

## Discussion

The emergence of viruses resistant to anti-SARS-CoV-2 drugs is a public health concern. Indeed, monoclonal antibody therapies face increasing resistance from viruses with amino acid substitutions in

their spike protein. In contrast, antiviral compounds targeting virus proteins other than the spike protein are becoming more prevalent and important. However, amino acid substitutions that reduce virus sensitivity to such compounds have been found in viruses isolated from COVID-19 patients[8–11] and in laboratory experiments[14–18,22,23].

Here, we attempted to characterize mutant viruses carrying amino acid substitutions linked to resistance to the oral antiviral nirmatrelvir, which targets 3CLpro (nsp5), since it is the antiviral drug used most often to treat COVID-19[18]. Of the nirmatrelvir-resistant substitutions reported, we chose Nsp5-L50F/E166V[19] and Nsp5-L50F/E166A/L167F[20] because they appear to maintain virus fitness in vitro. As expected, both mutant viruses reached a similar virus titer to that of the wild-type virus and were not inhibited by nirmatrelvir, although virus growth in cultured cells was delayed. In the hamster infection model, the two resistant viruses were slightly attenuated, maintained airborne transmissibility with slightly reduced efficiency, and were unable to compete with the growth of the wild-type virus in the absence of nirmatrelvir; however, although Nsp5-L50F/E166V was dominated by the wild-type virus in the nasal turbinate in the presence of the drug, it dominated in three of five animals in the lungs. A previous study showed that Nsp5-L50F/E166A/L167F replicates in and damages lungs at similar levels to wild-type virus and directly transmits from infected hamsters to naïve

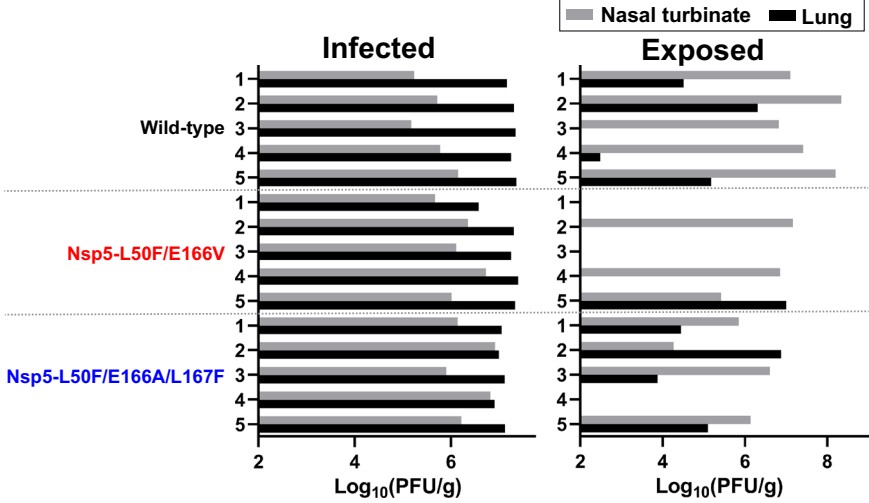

**Fig. 4 | Transmission of Nsp5-L50F/E166V and Nsp5-L50F/E166A/L167F between hamsters.** Hamsters were infected with the indicated virus at 10⁵ PFU. One day after infection, naïve hamsters were exposed to the infected hamsters. The infected hamsters (left) and the exposed hamsters (right) were euthanized at 4 days post-infection or 3 days post-exposure. Nasal turbinates (gray bars) and lungs (black bars) were collected for virus titration on VeroE6/TMPRSS2 cells. A two-sided Fisher's exact test showed that the transmission efficiency between hamsters was not statistically different among the three groups.

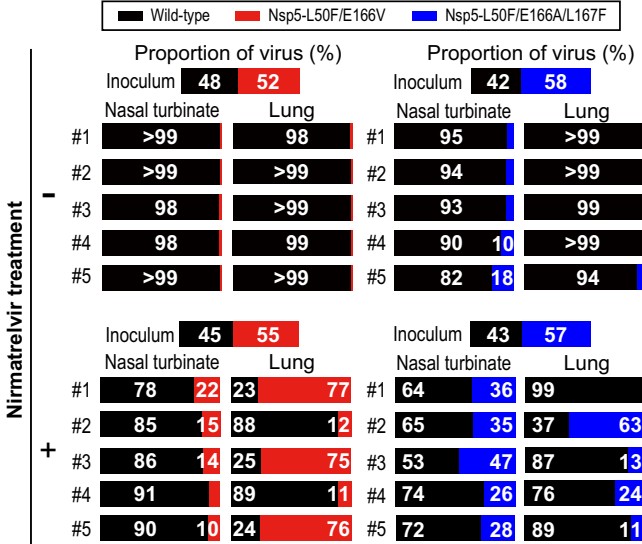

**Fig. 5 | Growth competition between Nsp5-L50F/E166V or Nsp5-L50F/E166A/ L167F and wild-type virus in hamsters.** Wild-type virus and Nsp5-L50F/E166V (left) or Nsp5-L50F/E166A/L167F (right) were mixed at a ratio of 3:7 based on their virus titers, and the virus mixture was intranasally inoculated into hamsters ($n = 10$ per group). Hamsters were treated with nirmatrelvir ($n = 5$, lower panels) or left untreated ($n = 5$, upper panels). Nasal turbinates and lungs were collected from the infected animals at 4 dpi and the frequency of each virus was determined by deep sequence analysis. The frequency of wild-type virus in the treated hamsters was compared with that in the untreated animals by using the Mann–Whitney test (two-sided) followed by the two-stage step-up procedure of the Benjamini, Krieger, and Yekutieli test. The frequency of wild-type virus in the nasal turbinate and lungs of treated hamsters was significantly decreased in co-infection experiments with Nsp5-L50F/E166V ($p = 0.00794$ and $p = 0.00794$) or Nsp5-L50F/E166A/L167F ($p = 0.00794$ and $p = 0.0476$), respectively.

hamsters[21]. Thus, Nsp5-L50F/E166V and Nsp5-L50F/E166A/L167F show similar or slightly reduced pathogenicity in hamsters. However, even though these nirmatrelvir-resistant mutants emerge, wild-type viruses outcompete them in the absence of nirmatrelvir, whereas in the presence of the drug, Nsp5-L50F/E166V may outcompete the wild-type virus in the lungs to some extent.

Viruses, especially RNA viruses, generate variants. Although SARS-CoV-2 has a proofreading mechanism, many people have been infected and each individual producing a large amount of virus could lead to the emergence of viruses resistant to monoclonal antibodies or antivirals when used as monotherapy. In influenza patients, viruses resistant to neuraminidase (NA) or PA inhibitors have been detected after treatment with the antivirals[24]. Some of these drug-resistant viruses are less fit than the wild-type virus[25], but others are not[26–28]. Regardless of whether viral fitness is reduced, such resistant viruses rarely spread worldwide. If a resistant virus does spread globally, either the selective pressure is extremely high due to the widespread use of the antiviral (which is unlikely), or the resistant virus acquires greater fitness through the acquisition of additional amino acid substitutions[25,29]. Combination therapy is thought to be a better strategy to treat viral infections and reduce the likelihood of emergence of resistant viruses[30], but it is possible for viruses resistant to both antivirals to emerge[31,32].

In conclusion, Nsp5-L50F/E166V and Nsp5-L50F/E166A/L167F are unlikely to dominate in nature. However, secondary adaptive mutations may make these mutant viruses dominant to wild-type virus. Moreover, mutant viruses possessing other resistance-conferring substitutions could emerge and dominate in the human population. Therefore, we need to closely monitor the amino acid mutations that confer resistance to nirmatrelvir and secondary adaptive mutations by undertaking active virus surveillance to detect such viruses as soon as possible after emergence.

## Methods
### Ethics
All animal experiments were conducted in accordance with the University of Tokyo's Regulations for Animal Care and Use, which were approved by the Animal Experiment Committee of the Institute of Medical Science, the University of Tokyo. The committee acknowledged and accepted both the legal and ethical responsibility for the animals, as specified in the Fundamental Guidelines for Proper Conduct of Animal Experiment and Related Activities in Academic Research Institutions under the jurisdiction of the Ministry of Education, Culture, Sports, Science, and Technology of Japan.

All experiments with SARS-CoV-2 were performed in enhanced biosafety level 3 (BSL3) containment laboratories at the University of

Tokyo, which are approved for such use by the Ministry of Agriculture, Forestry, and Fisheries, Japan.

## Cells

VeroE6/TMPRSS2 (JCRB 1819) cells were propagated in the presence of 1 mg/ml geneticin (G418; Invivogen) and 5 μg/ml plasmocin prophylactic (Invivogen) in Dulbecco's modified Eagle's medium (DMEM) containing 10% FBS. VeroE6-TMPRSS2-T2A-ACE2 cells were cultured in DMEM containing 10% Fetal Bovine Serum (FBS), 100 U/mL penicillin–streptomycin, and 10 μg/mL puromycin. HEK293T cells were cultured in Dulbecco's modified Eagle medium supplemented with 10% FBS. All cells were maintained at 37 °C with 5% $CO_2$. The cells were regularly tested for mycoplasma contamination by using PCR and confirmed to be mycoplasma free.

## Antivirals

Active components of remdesivir and molnupiravir (i.e., GS-441524 and EIDD-1931), and nirmatrelvir (PF-07321332) were purchased from MedChemExpress. S-217622 (ensitrelvir) was kindly provided by Shionogi Co., Ltd. Compounds were dissolved in dimethyl sulfoxide (in vitro) or 0.5% methylcellulose (in vivo) prior to use.

## Viruses

The full-genome nucleotide sequence of SARS-CoV-2 (hCoV-19/USA/WI-UW-5250/2021, B.1.617.2) was assembled into the pBeloBAC11 vector to generate infectious cDNA clones under the control of a cytomegalovirus (CMV) promoter as described previously[33,34]. The mutations responsible for the L50F/E166V and L50F/E166A/L167F substitutions in 3CLpro were introduced during the PCR step. To rescue these viruses, pBeloBAC11 encoding the wild-type virus [hCoV-19/USA/WI-UW-5250/2021 (B.1.617.2, delta variant), GenBank Accession No. OR116091], Nsp5-L50F/E166V (Accession No. OR116092), or Nsp5-L50F/E166A/L167F (Accession No. OR116093) was transfected into HEK293T cells. At 3 days post-transfection, the supernatant containing the viruses was collected and inoculated onto VeroE6/TMPRSS2 at 37 °C to prepare virus stocks. The stock viruses were deep sequenced to confirm the absence of unwanted mutations, and no position contained unwanted nucleotides that exceeded 10% of the population in all stock viruses.

## Deep sequence analysis

The whole genome of SARS-CoV-2 was amplified by using a modified ARTIC network protocol in which some primers were replaced or added[35]. In brief, viral RNA was extracted by using a QIAamp Viral RNA Mini Kit (QIAGEN). cDNA was synthesized by using a Lunar-Script RT SuperMix Kit (New England BioLabs) and subjected to a multiplexed PCR in two pools using ARTIC-N1 primers v5 and Q5 Hot Start DNA polymerase (New England BioLabs). The DNA libraries for Illumina NGS were prepared from pooled amplicons by using a QIAseq FX DNA Library Kit (QIAGEN) and then analyzed by using the iSeq100 System in 150-bp paired-end mode using an iSeq 100 i1 Reagent v2 (300-cycle) kit (Illumina). The reads were assembled by CLC Genomics Workbench (version 22, Qiagen). The average coverage depths of the wild-type virus, L50F/E166V, and L50F/E166A/L167F were 8,078×, 10,742×, and 8,731×, respectively.

## Focus reduction assay

Antiviral susceptibilities were determined by using a focus reduction assay as previously reported[2–4,36]. Briefly, VeroE6-TMPRSS2-T2A-ACE2 cells in 96-well plates were infected with SARS-CoV-2 at 100–400 focus forming unit/well. After a 1-h incubation at 37 °C, the inoculum was replaced with 1% Methyl Cellulose 400 (FUJIFILM Wako Pure Chemical Corporation) in culture medium containing serial dilutions of the antiviral compounds. The cells were incubated for 18 h at 37 °C and then fixed with formalin. The cells were stained with a mouse monoclonal antibody against SARS-CoV-2 nucleoprotein, clone N45 (TAUNS Laboratories, Inc., Japan), followed by a horseradish peroxidase-labeled goat anti-mouse immunoglobulin (Jackson ImmunoResearch Laboratories Inc.). Foci were visualized by using TrueBlue Substrate (SeraCare Life Sciences). The focus numbers were quantified by using an ImmunoSpot S6 Analyzer, ImmunoCapture software, and BioSpot software (Cellular Technology). The 50% inhibitory concentration ($IC_{50}$) values and 95% confidence intervals were calculated by using GraphPad Prism 9.3.0 (GraphPad Software).

## Growth kinetics in vitro

VeroE6/TMPRSS2 cells were infected with the indicated virus at a multiplicity of infection (MOI) of 0.001. After a 1-h incubation at 37 °C, the inoculum was replaced with medium ± 20 μM nirmatrelvir. Cell culture supernatants were collected at 8, 16, 24, 32, 40, 48, 56, and 72 h post-infection. Virus titers were determined by the use of a plaque assay in VeroE6/TMPRSS2 cells.

## Experimental infection of Syrian hamsters

Five- to six-week-old male Syrian hamsters (Japan SLC) were used in this study. Since SARS-CoV-2 occasionally fails to replicate in female hamsters, we used only male hamsters in this study. Under isoflurane anesthesia, five hamsters per group were intranasally inoculated with $10^5$ plaque forming unit (PFU) of the indicated virus. Body weights were measured daily before inoculation and 10 days after inoculation. Respiratory parameters [Penh (a nonspecific assessment of breathing patterns) and Rpef (a measure of airway obstruction)] were also measured by using a whole-body plethysmography system (PrimeBioscience) as previously described[37,38].

For the virus titration, five hamsters per group were intranasally infected with $10^5$ PFU of the indicated virus. At 3 and 6 days post-infection, the animals were euthanized and nasal turbinates and lungs were collected. The virus titers in these organs were determined by use of plaque assays on VeroE6/TMPRSS2 cells.

The transmission study was performed as previously described[39,40]. Briefly, five hamsters per group were intranasally infected with $10^5$ PFU of each indicated virus and placed in cages. One day after infection, naïve hamsters were placed in adjacent cages, 5-cm apart, separated by a double-layer partition with unidirectional air flow. Infected and exposed hamsters were euthanized at 4 days post-infection or 3 days after exposure. Nasal turbinates and lungs were collected for virus titration on VeroE6/TMPRSS2 cells.

For the co-infection study, wild-type virus and Nsp5-L50F/E166V or Nsp5-L50F/E166A/L167F were mixed at a ratio of 3:7 based on their titers, and the virus mixture (total $2 \times 10^5$ PFU) was inoculated into ten hamsters per group. One day after inoculation, five hamsters per group were treated orally with nirmatrelvir at 250 mg/kg twice daily as previously described[41,42]. The remaining five hamsters per group were left untreated. The animals were euthanized at 4 dpi and their nasal turbinates and lungs were collected for deep sequence analysis. The ratio of wild-type to Nsp5-L50F/E166V or Nsp5-L50F/E166A/L167F was calculated from the differences at positions 10202 and 10551 or 10202, 10551, and 10555, respectively. Samples with read depths of more than 1000 were analyzed.

## Statistical analysis

GraphPad Prism software version 9.3.0 was used to calculate P values. We compared virus titers in hamsters with the control by using a one-way ANOVA followed by Tukey's multiple comparisons. Body weight and virus growth kinetics were compared by using a two-way ANOVA followed by Tukey's multiple comparisons. Transmission efficiency was compared by using a two-sided Fisher's exact test. The frequency of wild-type virus in the nasal turbinate and lungs of the treated hamsters was compared with that of the untreated hamsters by using the Mann–Whitney test followed by the two-stage step-up procedure

of the Benjamini, Krieger, and Yekutieli test. Differences between groups were considered significant for *P* values < 0.05.

## Reagent availability
All reagents described in this paper are available through Material Transfer Agreements.

## Reporting summary
Further information on research design is available in the Nature Portfolio Reporting Summary linked to this article.

## Data availability
All data supporting the findings of this study are available within the paper and are provided in the Source data file. There are no restrictions to obtaining access to the primary data. The virus genome sequences of the wild-type virus, Nsp5-L50F/E166V, and Nsp5-L50F/E166A/L167F are available in GenBank (Accession Nos. OR116091, OR116092, and OR116093, respectively).

## Code availability
No code was used during the data acquisition or analysis.

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

## Acknowledgements
The authors thank Susan Watson for editing the manuscript.

## Author contributions
M.K., S.Y., and Y.K. designed the study. M.K., Y.F., R.U., M.I., and S.Y. performed the experiments. M.K., S.Y., and Y.K. analyzed the data and wrote the manuscript. All authors reviewed the manuscript and approved the final version.

## Funding
This work was supported by the Japan Agency for Medical Research and Development (JP22wm0125002 and JP223fa627001).

## Competing interests
Y.K. has received unrelated funding support from Daiichi Sankyo Pharmaceutical, Toyama Chemical, Tauns Laboratories, Inc., Shionogi & Co. LTD, Otsuka Pharmaceutical, and KM Biologics. The other authors have no conflicts of interest.
