## [Peer Review File · Nature Communications]

REVIEWER COMMENTS

Reviewer #1 (Remarks to the Author):

Kisa et al. engineered two sets of mutations, which confer resistance to MPro drugs, in the backbone of Delta variant SARS-CoV-2. They showed the mutant viruses were less fit than wildtype virus in hamsters in a competition assay. Although the research topic is important and the experiments were well performed, the novelty of the study is not high. A number of previous studies have reported the resistance mutations. The current study just used these mutations to analyze their effect on viral fitness in cell culture and hamsters.

Major comments

1. The novelty of the study is not high.
2. Have the authors experimentally tested the hypothesis that L50F compensates the loss of viral fitness caused by E166 and L167 mutations?

Reviewer #2 (Remarks to the Author):

Paxlovid resistance is of significant public health importance as it is the best/only current outpatient therapy for COVID. In the manuscript by Kiso et al the authors characterize two combinations of mutations that confer SARS-CoV-2 resistance to nirmatrelvir which is the protease inhibitor component of paxlovid. These mutations were L50F/E166V and L50F/E166A/L167F in Mpro/Nsp5 in a delta virus background. They test these viruses in vitro and in vivo and demonstrate that while they are resistant to paxlovid in vitro, they have impaired in vitro replication in the absence of paxlovid and decreased pathogenesis in hamsters. The study is technically well done with careful viral growth curves and appropriate use of a hamster model but the biological insight and innovation are limited. In this reviewer's opinion, the study is premature as it is limited in scope and offers minimal new mechanistic insight. Additional experimentation as described below would be of value.

Major:

- 1) The competition experiments in figure 3 should be repeated in the presence of paxlovid.
- 2) The study assesses viral replication and pathogenesis but transmission may be the most important phenotype if the concern is that widespread paxlovid use will drive resistance. The authors should leverage their expertise to see if there is a defect in transmission from infected hamsters to cagemates.

3) The role of each individual mutation is unclear. Is the entire phenotype mediated by L50F or E166 for instance? This is important for optimizing surveillance and mechanistic insight.

4) The study conflicts with recent preprints (ie Abdelnabi et al which show that one of these viruses exhibits similar pathogenesis in hamsters). How do the authors reconcile these differences?

Minor:

5) The relevance of these mutation combinations is a bit unclear. Have these mutant combinations every been identified in primary isolates from patients?

6) Only male animals used. Both sexes should be used unless a clear rationale is given.

7) Table 1: It would be useful to show the curves for these data and state the confidence interval for the IC50 calculations.

8) The terms Penh and Rpef should be defined. This terminology is not standard to most readers.

Reviewer #3 (Remarks to the Author):

This is an interesting study monitoring the emergence of nirmatrelvir-resistance SARS-CoV-2 variants in animal model and VeroE6/TMPRSS2 cells susceptible to be infected with the CoV. The study is well conducted using cell culture and also an animal model and perform a study of competition between wild type virus and mutants to study fitness of resistant mutations inserted in Delta variant. It is nice to read how authors correctly use the terms “resistance” mutations and “resistant” mutants, and despite the manuscript is clear and understandable, I would recommend to correct the English by an English native.

I would recommend publication after minor changes.

Minor comments:

Lines 38 to 39: Despite viruses carrying these mutations have a lower fitness than wild type, nobody can assure that they will have the capability to dominate in nature, since compensatory mutations might arise, as has been shown in many other viral infections.

Methods lines 168: Authors specify that they mix the wild type (wt) with each mutant at a ratio 7:3. Later on, in results line 244, the ratio is 3:7 because now they are using the order mutant:wt. In Fig 3 the order is again wt:mutant. I wonder if it is possible for the authors to keep the same order along the text. It will facilitate the reading.

Line 124-125: To confirm the absence of unwanted mutations the stock of viruses were subjected to deep-sequencing. How deep? which was the coverage? For a virus such as SARS-CoV-2 with low mutations rate at least 10000 reads and ideally more than 30000 should be performed.

Line 134-135: iSeq100 system (Illumina), please provide more data on the protocol used, length of the reads obtained (kit used) 1x36, 1x50, 1x75, 2x74 or 2x150pb?

Line 137-139: Citations of "Focus reduction method". Please also quote the original paper that describe this method: "Vanderheiden A, Edara VV, Floyd K, et al. Development of a Rapid Focus Reduction Neutralization Test Assay for Measuring SARS-CoV-2 Neutralizing Antibodies. *Current Protocols in Immunology*. 2020;131:e116".

Discussion

I would recommend to enrich the discussion on the problem of treating viral infections using monotherapy regimens, inserting something like: One of the greatest problem when treating viral infections, is the use of antiviral treatments in monotherapy. Viruses, and especially RNA viruses, are entities able to generate variation. SARS-CoV-2 despite having a proofreading mechanism, it is highly probable that with the large viral loads that are reaching in every infection (10¹¹ during the peak of infection), and the large number of people infected, a resistant virus to a monoclonal antibody or to any antiviral used in monotherapy is selected. The best strategy to fight against viral infections is the combination therapy as firstly reported by Domingo E in 1989 (Domingo E. (1989) RNA virus evolution and the control of viral disease. *Prog Drug Res*. 33:93-133. https://doi.org/10.1007/978-3-0348-9146-2_5). Please quote this paper.

The manuscript by Kiso et al also focus in another very interesting point, which is the selection of resistance mutations that in many cases can compromise fitness. I would recommend to enrich discussion on this topic: It has been described that selection of resistance associated substitutions, even using combination therapies (Chen Q, Perales C, et al Deep-sequencing reveals broad subtype-specific HCV resistance mutations associated with treatment failure. (2020) *Antiviral Res*. 174:104694.),

can compromise viral fitness when competing with wild type in the absence of the antiviral (RNA virus mutations and fitness for survival. Domingo E, Holland JJ. *Annu Rev Microbiol*. 1997;51:151-78. doi: 10.1146/annurev.micro.51.1.151), however, other mutations appearing in the same genome might compensate the loss of fitness and the mutant virus can outcompete wild type and become the dominant variant (Donaldson EF et al Clinical evidence and bioinformatics characterization of potential hepatitis C virus resistance pathways for sofosbuvir. *Hepatology*. 2015 Jan;61(1):56-65. doi: 10.1002/hep.27375.).

Reviewer #1 (Remarks to the Author):

Major comments

1. The novelty of the study is not high.

Although many papers have reported on amino acid substitutions that reduce susceptibility to nirmatrelvir, there are no reports that have analyzed the pathogenicity of nirmatrelvir-resistant viruses in hamsters in detail. Moreover, competitive infection experiments between drug-resistant and wild-type viruses and airborne transmission experiments between hamsters have provided useful information about the possible spread of resistant viruses. Therefore, we believe this manuscript includes novel information.

2. Have the authors experimentally tested the hypothesis that L50F compensates the loss of viral fitness caused by E166 and L167 mutations?

The main aim of this study is to investigate the pathogenicity of resistant viruses and their fitness over the wild-type virus. The compensation by the L50F substitution is not the focus of this study. Furthermore, Jockmans et al. already investigated the role of the L50F substitution in combination with the E166A and L167F substitutions and it was not clear whether L50F compensates for the loss of viral fitness caused by the E166 and L167 mutations. Please see reference #20 (Jochmans D et al. The Substitutions L50F, E166A, and L167F in SARS-CoV-2 3CLpro Are Selected by a Protease Inhibitor In Vitro and Confer Resistance To Nirmatrelvir. 2023. mBio 14:e0281522.).

Reviewer #2 (Remarks to the Author):

Major:

1) *The competition experiments in figure 3 should be repeated in the presence of paxlovid.*

In response to the reviewer's comment, we performed the competition experiments in the presence of nirmatrelvir. We found that one of the drug-resistant viruses dominated in the lungs of three of five animals, and that another drug-resistant virus did the same in the lungs of one of five animals. However, in the nasal turbinate, the wild-type virus dominated at the end of the experimental period. These results are now included in Figure 5.

2) *The study assess viral replication and pathogenesis but transmission may be the most important phenotype if the concern is that widespread paxlovid use will drive resistance. The authors should leverage their expertise to see if there is a defect in transmission from infected hamsters to cagemates.*

In response to the reviewer's comment, we performed the airborne transmission experiment between hamsters. The experiment showed that both mutant viruses maintained airborne transmissibility between hamsters, although the transmissibility might be slightly reduced compared with that of the wild-type virus. These results are now included in Figure 4.

3) *The role of each individual mutation is unclear. Is the entire phenotype mediated by L50F or E166 for instance? This is important for optimizing surveillance and mechanistic insight.*

In response to the reviewer's comments, we have added the information to the main text (Page 4, lines 80–83).

4) *The study conflicts with recent preprints (ie Abdelnabi et al which show that one of these viruses exhibits similar pathogenesis in hamsters). How do the authors reconcile these differences?*

In the manuscript by Abdelnabi et al., hamsters were infected with 10^4 TCID₅₀ of Wuhan-type virus or its mutant that possesses L50F-E166A-L167F. In

contrast, we infected hamsters with 10^5 PFU of delta variant and its mutant that possesses the identical mutations. These differences (infection dose and backbone virus) might have caused the difference between the two studies.

Minor:

5) The relevance of these mutation combinations is a bit unclear. Have these mutant combinations every been identified in primary isolates from patients?

These mutations were identified in viruses selected in vitro. This point is now included in the main text (Page 4, lines 76–81).

6) Only male animals used. Both sexes should be used unless a clear rationale is given.

Although we do not know the exact reason, SARS-CoV-2 occasionally fails to replicate in female hamsters. Therefore, we used only male hamsters.

7) Table 1: It would be useful to show the curves for these data and state the confidence interval for the IC50 calculations.

In response to the reviewer's comment, we have changed Table 1 to Figure 1 to show the dose-response curves and 95% confidence intervals.

8) The terms Penh and Rpef should be defined. This terminology is not standard to most readers.

We have added explanations to the main text (Page 7, lines 168–169).

Reviewer #3 (Remarks to the Author):

It is nice to read how authors correctly use the terms “resistance” mutations and “resistant” mutants, and despite the manuscript is clear and understandable, I would recommend to correct the English by an English native.

The manuscript has been edited by a native-English-speaking professional editor.

Minor comments:

Lines 38 to 39: Despite viruses carrying these mutations have a lower fitness than wild type, nobody can assure that they will have the capability to dominate in nature, since compensatory mutations might arise, as has been shown in many other viral infections.

In response to the reviewer’s comment, we have modified the last sentence of the abstract (Page 2, lines 42–43).

Methods lines 168: Authors specify that they mix the wild type (wt) with each mutant at a ratio 7:3. Later on, in results line 244, the ratio is 3:7 because now they are using the order mutant:wt. In Fig 3 the order is again wt:mutant. I wonder if it is possible for the authors to keep the same order along the text. It will facilitate the reading.

In response to the reviewer’s comment, we have changed the text to be consistent in both places (Page 12, lines 276–277).

Line 124-125: To confirm the absence of unwanted mutations the stock of viruses were subjected to deep-sequencing. How deep? which was the coverage? For a virus such as SARS-CoV-2 with low mutations rate at least 10000 reads and ideally more than 30000 should be performed.

We believe that a minimum depth was achieved because the average coverage depths of the wild-type virus, L50F/E166V, and L50F/E166A/L167F were 8,078×, 10,742×, and 8,731×, respectively. This point is now included in the text (Page 6, lines 140–142).

Line 134-135: iSeq100 system (Illumina), please provide more data on the protocol used, length of the reads obtained (kit used) 1x36, 1x50, 1x75, 2x74 o 2x150pb?

We used an iSeq 100 i1 Reagent v2 (300-cycle) kit. This point is now included in the text (Page 6, lines 138–139).

Line 137-139: Citations of “Focus reduction method”. Please also quote the original paper that describe this method: “Vanderheiden A, Edara VV, Floyd K, et al. Development of a Rapid Focus Reduction Neutralization Test Assay for Measuring SARS-CoV-2 Neutralizing Antibodies. Current Protocols in Immunology. 2020;131:e116”.

In response to the reviewer’s comment, we have added the suggested reference as reference #24.

Discussion

I would recommend to enrich the discussion on the problem of treating viral infections using monotherapy regimens, inserting something like: One of the greatest problem when treating viral infections, is the use of antiviral treatments in monotherapy. Viruses, and especially RNA viruses, are entities able to generate variation. SARS-CoV-2 despite having a proofreading mechanism, it is highly probable that with the large viral loads that are reaching in every infection (10¹¹ during the pick of infection), and the large number of people infected, a resistant virus to a monoclonal antibody or to any antiviral used in monotherapy is selected. The best strategy to fight against viral infections is the combination therapy as firstly reported by Domingo E in 1989 (Domingo E. (1989) RNA virus evolution and the control of viral disease. Prog Drug Res. 33:93-133. https://doi.org/10.1007/978-3-0348-9146-2_5). Please quote this paper. The manuscript by Kiso et al also focus in another very interesting point, which is the selection of resistance mutations that in many cases can compromise fitness. I would recommend to enrich discussion on this topic: It has been described that selection of resistance associated substitutions, even using combination therapies (Chen Q, Perales C, et al Deep-sequencing reveals broad subtype-specific HCV resistance mutations associated with treatment failure. (2020) Antiviral Res. 174:104694.), can compromise viral fitness when competing with wild type in the absence of the antiviral (RNA virus mutations and fitness for survival. Domingo E, Holland JJ. Annu Rev Microbiol. 1997;51:151-78. doi: 10.1146/annurev.micro.51.1.151), however, other mutations appearing in the same genome might compensate the loss of fitness and the mutant virus can outcompete wild type and become the dominant variant (Donaldson EF et al Clinical evidence and bioinformatics characterization of potential hepatitis C

virus resistance pathways for sofosbuvir. Hepatology. 2015 Jan;61(1):56-65. doi: 10.1002/hep.27375.

In response to the reviewer's comment, we now discuss this point (Page 14, line 320–Page 15, line 331).

REVIEWER COMMENTS

Reviewer #2 (Remarks to the Author):

The revised manuscript is significantly improved.

Statistics should be added to the new figure 4 and figure 5 data to prove that the transmission defect is statistically significant. This is important for the papers conclusions and the impact of the study. 5 hamsters per group is not particularly rigorous and if the differences are not significant a larger sample size is needed to make a definitive conclusion.

The authors should add a line to the methods explaining why only male hamsters were used (reviewer #2 point 6).

Reviewer #2 (Remarks to the Author):

Statistics should be added to the new figure 4 and figure 5 data to prove that the transmission defect is statistically significant. This is important for the papers conclusions and the impact of the study. 5 hamsters per group is not particularly rigorous and if the differences are not significant a larger sample size is needed to make a definitive conclusion.

In response to the reviewer's comment, we conducted statistical analyses: a two-sided Fisher's exact test for the transmission study, and the Mann-Whitney test followed by the two-stage step-up procedure of the Benjamini, Krieger, and Yekutieli test for the competitive study. These analyses revealed that the transmission efficiency was not significantly different between the wild-type virus and mutant viruses. We also found that the frequency of wild-type virus was significantly decreased in the nasal turbinate and lungs of treated hamsters compared to untreated hamsters. This information has been added to the text.

The authors should add a line to the methods explaining why only male hamsters were used (reviewer #2 point 6).

In response to the reviewer's comment, we have added an explanation to the method section (Page 7, lines 165–166).